# Digitalization in Developing Maritime Business Environments towards Ensuring Sustainability

**Nexhat Kapidani** [1], **Sanja Bauk** [2,*] **and Innocent E. Davidson** [3]

[1]   Administration for Maritime Safety and Port Management of Montenegro, 85000 Bar, Montenegro; nexhat.kapidani@pomorstvo.me

[2]   Maritime Studies Department, Durban University of Technology, Durban 4000, South Africa

[3]   Department of Power Engineering, Durban University of Technology, Durban 4000, South Africa; innocentd@dut.ac.za

*   Correspondence: sanjab@dut.ac.za; Tel.: +27-031-373-2694

**Abstract:** The paper focuses on assessing the level of digitalization in several developing maritime business environments in Albania, Bosnia and Herzegovina, Montenegro, and Serbia. The assessment has been done in reference to Holtham's and Courtney's Intelligent Information and Communication Technologies (ICT) Exploiter Model. The dimensions as maritime business system effectiveness, roles, and skills of information technology personnel, ladders of knowledge, ICT strategy, organizational culture, and manager's mindset are analyzed. In addition, benchmarking with findings from developed maritime business environments in Croatia, Greece, Italy, and Slovenia, which belong to the European Union (EU), by using the same model, has been conducted. This is done with the aim to outline directions for improving the quality and speed of digitalization in non-EU countries, which have been functioning for decades in transitional conditions. The maritime ecosystem naturally has a tendency to be unique and to function smoothly as such. Alleviating the differences in the level and effectiveness of digitalization in developed and developing European countries is a path towards achieving this goal. By sharing their own expertise in the rational and intelligent use of ICT, developed EU countries can support developing non-EU countries towards ensuring sustainability in the entire European and worldwide maritime business ecosystem.

**Keywords:** digitalization; maritime; business; intelligent use of ICT; sustainability

## 1. Introduction

The term maritime deals with the world's oceans. The world's oceans belong to everyone and are an essential life support ecosystem. They absorb carbon dioxide from the atmosphere, generate up to half of the world's oxygen supply, provide essential protein for nearly three billion people, regulate global climate, provide numerous resources used by humans, etc. They enable performing more than 90% by volume and 70% by value of world trade [1] since sea transportation is still the most-effective way of transporting raw materials and goods around the globe.

On the other side, marine habitats, nearshore ecosystems and coastal communities face huge pressures that threaten their sustainability through climate change, ocean acidification, rising sea level, variable fish stock, natural and human-caused disasters, and the like [2].

A paradox related to maritime is the lack of digitalization in today's digital age compared to other spheres of human lives and business activities. A recent systematic literature review of major journal databases resulted in finding only 99 research papers in maritime on topics such as autonomous vehicles and robotics, artificial intelligence, big data, virtual reality, augmented and mixed reality, the Internet of Things, the cloud and edge computing, digital security, 3D printing, and additive

engineering, across ship design and shipbuilding, shipping, and ports areas [3]. The same authors stated: "Although maritime transport is the backbone of world commerce, its digitalization lags significantly behind when we consider some basic facts." According to the same source, these facts are that inter-organizational information systems (IOS) are used 75% in the hinterland and only 25% at maritime; ports lag behind in regard to the utilization of information technologies/information systems (IT/IS) [4]; the EU lacks a clear strategy toward digitalization in the maritime industry [5]; basic security is still more of an aim than a reality, even with the tools of the Internet of Things (IoT), which are as widely-used as radio frequency identification (RFID), where it is difficult to find commercial systems without critical security flaws and vulnerabilities [6]; maritime cloud, which is still in its fledgling stages [7], etc. When it comes to digitalization in maritime logistics, Fruth and Teuteberg [8] found out that it is still in its initial stage. This wide-ranging search revealed only a small number of scientific literatures and showed that digitization in the maritime logistics chain is currently being addressed and considered in practical rather than scientific literature. In addition, Fernardo [9] stated that information technology is surprisingly neglected in maritime literature.

Some tangible examples speak in favor of the above theoretical findings. For instance, there is only one autonomous operational ferry boat, "Falco", built by Rolls-Royce and Finferries, while another one, "YARA Birkeland", is planned to be completely operational by the year 2022 [10]. On the contrary, 1400 self-driving vehicles are tested on roads by more than eighty United States of America companies, plus 1.59 million registered drones are in operation [11]. Furthermore, containers with radioactive materials are tracked by RFID technology at the level of any single item/container on road and rail transportation [12,13], but not in maritime [14]. To the best of the authors' knowledge, tracking of containers at the unit level after they leave the departure port and until they reach the arrival port has not yet been achieved. It seems that the Internet of Things or everything is not as vivid at sea as it is on land. Maritime networks need performances close to high-speed terrestrial wireless broadband services on land, but there is a scarcity of in-depth research efforts in this domain [15]. The research of networks at sea is much more complex in comparison to land due to sea surface movements, wave occlusions, etc. Different communication channels, for instance, Very High-Frequency Data Exchange Service (VDES) and Navigation Data (NAVDAT), have been developing with the aim of overcoming the lack or low stability of internet connection at sea [16].

A non-negligible challenge when it comes to digitalization in maritime is the non-International Convention on Safety of Lives at Sea (non-SOLAS) ships. A few years ago, it was estimated that 60,000 non-SOLAS ships would use electronic navigation charts (ENC) and ENC up-date services [17]. The concept of smart or e-Navigation aims to reduce the number of accidents for SOLAS ships by 65%. However, non-SOLAS ships, due to their large numbers, should not be neglected, especially in confined waters and sea areas with heavy traffic. For instance, some experiments were done in Korea. Accidents of more than 3000 vessels were analyzed, and only 13% were SOLAS ships, while the rest were non-SOLAS ships [18]. Therefore, a large number of non-SOLAS ships will undoubtedly slow down the process of e-Navigation implementation.

Efforts are being made worldwide toward more intensive digitalization, both on-board and ashore. However, there are still many impediments on these paths caused by the differences among countries in terms of their economic development, including more or less complex inherent political, legal, and administrative barriers [19]. The scarcity of IT/IS research in developing countries, in general, and, in particular, in maritime, has also been revealed in the literature [20,21].

## 1.1. Problem Statement

Undoubtedly, there is room for further exploration when it comes to digitalization in maritime in developed, but, understandably, even more in developing countries. In this regard, the purpose of this study is to explore how and in what manner some developing non-EU maritime administration and business entities exploit available maritime info-communication systems and which components they use. Comparisons are made with reference to some EU countries in the same domain. As a theoretical

base, we used Holtham's and Courtney's Intelligent Information and Communication Technologies (ICT) Exploiter Model [22]. We aim to benchmark to what extent these entities use rationally available maritime info-communication systems and draw a path for improvements, in particular, when it comes to the sustainable development of the non-EU maritime sector. Improving the scope and level of ICT deployment in developed and developing countries is of crucial importance for providing sustainability in the maritime cluster and health relations between numerous players in this huge and complex business ecosystem in Europe and worldwide.

### 1.2. Sustainability Matter

Sustainability refers to the ability to exist constantly. In the 21st century, it generally refers to the capacity for the biosphere and human civilization to coexist [23]. It is about planning progress in the future without causing damage to the environment so as to guarantee a safe habitat to the next generations, who will continue to develop their economies and societies [24].

Sustainability is regarded as achieving economic, social, and environmental performances simultaneously that support an organization for long-term competitiveness [25]. In maritime, economic performance is manifested through a port's tendency to attract more freight and passengers; linear shipping companies striving to achieve a higher level of operational and financial effectiveness; maritime supply chain management endeavoring to synchronize processes and partners involved in achieving maximum profits, etc. Environmental sustainability has become a popular topic among academics and professionals in recent years, along with climate change concerns. It relates the ability of maritime companies to attain eco-efficiency in delivering services by reducing air emissions through vehicle speed and fuel consumption optimization, intensifying experiments with hydrogen-powered ships, monitoring and controlling effluent waste and hazardous materials, preventing environmental accidents, and developing contingency measures [9]. Social sustainability focuses on the needs of people and the requirements to implement corporate social responsibility. Being socially responsible benefits people and contributes to maritime companies' economic performance.

If today's focus is on developing smart (partly or fully autonomous) ships, then we have to think about smart ports and smart regulations in terms of preventing accidents in the ports and at sea by mitigating risks and prospective consequences. We have to also take into account the human dimension of technological growth and development in maritime by comparing the advantages and disadvantages of biological and virtual intelligence or virtual smartness. Furthermore, if we are developing e-Navigation, with the primary intention to reduce the number of accidents and reduce the ecological impact, then we have to think about non-SOLAS ships, as well. Additionally, as we are developing a completely connected global ecosystem, then we can not leave ships and ports as separate entities. It means we have to develop, implement and adopt smart IT/IS solutions in maritime as a whole. Within this context, smart IT/IS are those used rationally and intelligently in certain settings, including feasible plans for innovation success and progress. These systems should enable smarter collaboration, enhance operations, satisfy clients' expectations of transparency and predictability, and respond to societal concerns. They should increase the efficiency, safety, and ecological sustainability of the world's maritime industry [26].

Several comprehensive studies have been done in this direction. For instance, Lambrou et al. [27] addressed shipping incumbents' digitalization activities from both technological and management aspects. Sislian and Jaegler [28] analyzed how the port's use of Enterprise Resources Planning (ERP) systems affect the different perspectives of the sustainable maritime balanced scorecard in an efficient and effective manner. Furthermore, Fedi et al. [29] analyzed the influence of information technology solutions during the implementation of mandatory constraints in port operations regarding container verified gross mass to enhance maritime and port safety operations. The Port Community Systems (PCS) was found to have a positive influence on the adoption of this safety regulation. Lee and Nam [30] defined the concept of green shipping and analyzed key problems and countermeasures in its implementation. Felski and Zwolak [10] analyzed the ocean-going autonomous ship challenges and

threats at the example of the first fully autonomous small vessel "USV Maxlimer"—SEA-KIT type, which crossed from the UK to Belgium in 2019, etc.

Our research study is focused on managers' and employees' reflections on the optimal deployment of available IT/IS across different maritime organizations in several non-EU and EU countries, including the plans for further digitalization actions.

## 2. Methodological Framework

Within several previous studies in this field, we were focused on the design, implementation, adoption, and innovation success of contemporary ICT solutions in transitional environments [31–33]. We were driven by users' needs and preferences in some developing economies, faced with ongoing crises and constant lack of funds to provide up-to-date, comprehensive, sophisticated, and efficient ICT systems. For the purpose of this study, we used the Intelligent ICT Exploiter Model [22,34–36]. The Intelligent ICT Exploiter Model was developed upon several basic constructs connected with business entities and ways in which they conduct their businesses activities, i.e., knowledge, IT management, system efficiency through internal and external communications, organizational culture, ICT strategy, and top manager's mindset, which has to bind all other constructs intelligently. The model is shown in Figure 1. We also referred to respected IT experts, who claimed that success in the digital economy would be achieved by companies that are smart about how they use ICT [37]. Besides, we considered [38] a model of adopting new technologies in developing environments by considering innovation, economic, technological, usability, contextual, and organizational factors. Similar to this model is Keszey's [39], which considers environmental variables, organizational variables, easiness of IS use, perceived usefulness of IS, company type and tenure, with company ownership as a ruling variable. A comprehensive model has been developed by Sislian and Jaegler [40] to achieve green management objectives in the port, taking into consideration financial, internal, social, environmental, innovation, and workers' learning constructs. Our methodological framework was based on the triangulation of the previously mentioned approaches, focusing on Holtham's and Courtney's model. We consider the rationality of using available IT/IS systems of key importance in achieving business and prospective innovation implementation success in the considered developing non-EU countries. In fact, we assumed knowledge and skills as key perpetuators of success. Although we do not deny the importance of the previously mentioned factors, they require another type of research approach, and these factors (innovation, economic, technological, usability, contextual, and organizational) might be taken into consideration in further research work in the field. The proposed model was conceived at a high level of abstraction, and it applies to any administrative and business organization, including maritime ones.

### 2.1. Variables and Hypothesis

A theoretical framework involves the identification of a network of relationships among variables considered important to the problem. It is a foundation for the hypothetico–deductive approach, which we applied. After an extensive review of relevant literature sources in the field and developing a theoretical framework, primarily based on the Intelligent ICT Exploiter Model, we have drawn several hypotheses as tentative yet testable statements. These statements are, as usual, logically conjectured relationships between two or more variables expressed in the form of testable statements [41].

### 2.1.1. Independent Variables

Independent variables influence the dependent variable in either a positive or negative way. In our model, knowledge, IT management, effective maritime administrative or business system, and organizational culture are considered as the independent variables.

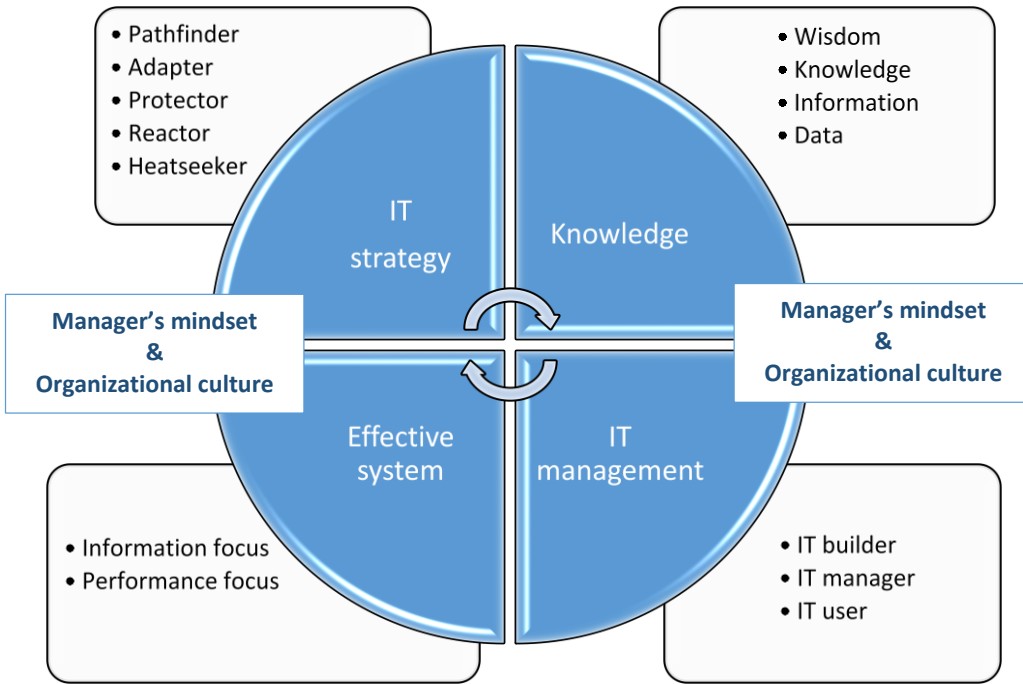

**Figure 1.** Intelligent Information and Communication Technologies (ICT) Exploiter Model (Adapted from: [22]).

*Knowledge.* It is metaphorically a stair of the knowledge ladder, including data, information, knowledge, and wisdom. There is a gap between the information technology revolution and the information revolution. The primary idea of the free and unlimited sharing of information failed since it did not take into account the commercial dimension of the process. Information is shared asymmetrically, and those who control the fastest and the biggest computers impose the rules [42]. Consequently, the path: data, information, knowledge, and wisdom is not an easy one, and it requires considerable effort towards achieving professional and business success. Most precisely, two vital tasks in modern enterprises are to speed up the creation of new knowledge by both individuals and communities and to accelerate the sharing of knowledge within and across communities. Due to this, we set the following hypothesis:

**Hypothesis 1 (H1).** *Knowledge is of key importance for the intelligent use of ICT.*

*IT Management.* It is based on IT builders or architects, IT managers, and IT users. A person or management team that communicates the needs of ICT users to IT builders or architects is present in the organization as a knowledge navigator or information resource manager. There are business organizations that recognized this triangle, and they are working on filling and improving all necessary skills in this direction [43,44]. Due to the triangle of roles and skills of IT builders, managers, and users, we set the hypothesis:

**Hypothesis 2 (H2).** *IT managers enable the intelligent use of ICT.*

*Effective system.* Such a system can be achieved by setting and communicating critical success factors (CSF) [45] and developing them steadily. The first step is to use technology to create an effective operational platform, primarily with internal information. Then, the CSFs can be widened to foster the improvement in skills in the use of technology. This will start with employees and then extend to suppliers and customers. When these two steps work well, the CSFs can be broadened to encompass external information about markets, customers, and competitors. After these three steps,



business intelligence allows organizations to identify and manage risk while developing new products, services, and markets to ensure a successful future. Upon this, we draw the following hypothesis:

**Hypothesis 3 (H3).** *Effective business systems manage and exploit both internal and external information.*

*Organizational culture.* While there is a universal agreement that it exists and plays an important role in shaping behavior in organizations, there is little consensus on what organizational culture actually is. We quoted several expressions that can be used in the absence of a universally accepted definition [46]: Organizational culture is how organizations do things; Organizational culture is the sum of values and rituals, which serve as the glue to integrate the members of the organization; Organizational culture is civilization in the workplace, etc. In this context, organizational culture is of particular importance since it permeates all considered constructs in a subtle way, and we analyzed it as the fourth construct in our model, connected with the hypothesis:

**Hypothesis 4 (H4).** *Positive organizational culture enhances the intelligent use of ICT.*

2.1.2. Moderating Variables

The moderating variables have a strong contingent effect on the independent variables–dependent variable relationship. In our study, the ICT strategy moderates this relationship.

*ICT Strategy.* It is a strategy that has to link business and technology. It has to ensure "C" for communication is fully integrated into strategic business thinking in both a technological and human sense. It is an assessment tool to assist organizations in identifying behavior regarding ICT adoption [47,48]. There are five strategic orientations, which are listed and explained below.

*Pathfinder*—Systematically seeks and selectively exploits relevant ICT trends to gain a competitive advantage and enable entry into new markets. The Pathfinder is willing to experiment with novel ICT. It constantly seeks a competitive advantage by detecting or sensing emerging ICT trends and possibilities.

*Adapter*—Operates in two types of market: one relatively stable and focused on efficiency, and the other where ICT plays an increasingly important role. The Adapter applies different rates of technological uptake in each. This split feature is typical for businesses where different technological uptake rates are applicable, and adaptability rather than uniform solutions are applicable.

*Protector*—Carefully evaluates ICT investment for its efficiency orientation and applies ICT primarily to reduce costs of investments and increase communication processes rather than market creation. The Protector is control orientated and slow to innovate. These organizations work in domains where core ICT-based technologies are universally available and easily replicable.

*Reactor*—Reactor is a characteristic of an organization where technology is not seen as a strategic tool. It responds slowly to change and tends to view ICT applications as standalone tools. In this strategic orientation, technology is not seen as being strategic, with ICT platforms often appearing to be very weak or obsolescent. The risk is that the organization could quickly become non-competitive through a lack of capacity to meet customers' needs.

*Heat seeker*—Sized upon ICT fashioned instead of strategically analyzing the best ICT fit for its business problems. Heat seeker is typical for an organization whose structure is in constant flux, moving to frequent new initiatives before obtaining sustained business performance. This organization is prone to frequent initiatives regarding ICT spends and subsequent partial reversals when intended benefits are not realized quickly.

We treated the ICT strategies as five moderating variables in our research model since they are influenced by previously considered constructs, and they have an impact on ICT intelligent exploitation. Upon the above stated, the following hypothesis can be drawn:

**Hypothesis 5 (H5).** *ICT strategies Pathfinder, Adapter, and Protector are "in fit", while Reactor and Heat Seeker are "out of fit" when it comes to intelligent ICT exploitation in an organization (Figure 2).*

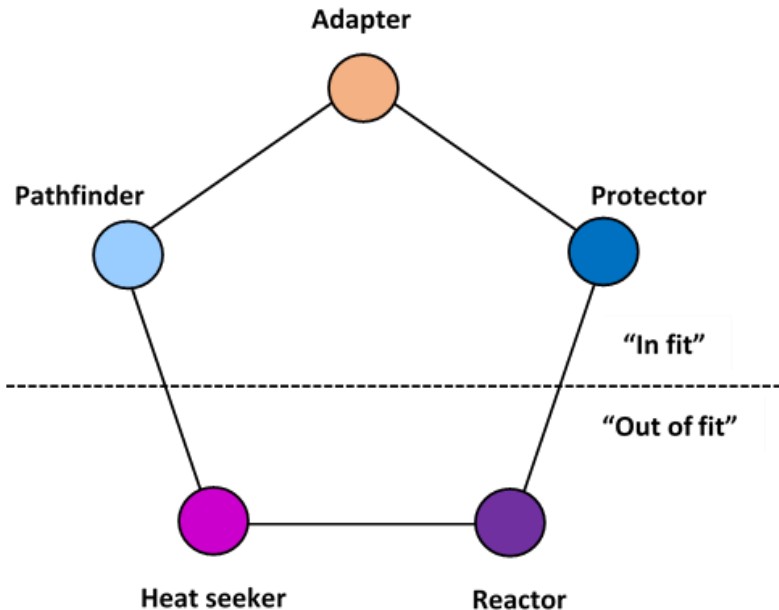

**Figure 2.** Five types of ICT strategic orientation (Source: Adapted from: [46]).

2.1.3. Mediating Variable

Bringing a mediating variable into play helps to model a process. The mediating variable surfaces as a function of the independent variables operating in any situation. It conceptualizes and explains the influence of independent variables on the dependent variable. In our model, the top manager's mindset is treated as a mediating variable.

*Mindset.* For the top manager or top management team, a metaphor can be used: Their role is to weave a fabric of horizontal (information, technology, people, and organization) and vertical (direction, knowledge, process, and climate) threads mutually intertwined. In organizations where knowledge is a core dimension, managers have frequently identified skills (people) as the major influence, commonly along with the organizational climate. Switching from an information-based to a knowledge-based enterprise is a major challenge for today's companies [49]. Therefore, managers have to combine notions from several different domains: organizational behavior, human resource management, artificial intelligence, IT/IS, etc. Technology is invariably cited as a key enabler, but not usually as significant overall as skills and climate. The top managers' team mindset covers all considered constructs, and it affects the dependent variable intelligent use of ICT. Therefore, the sixth hypotheses should be formulated as follows:

**Hypothesis 6 (H6).** *Manager's mindset is of crucial importance for the intelligent use of ICT.*

2.1.4. Control Variable

The control variables are an important part of an eye-tracking experiment. As a control variable in our research, we used non-compliance between technology-led potential and its everyday usage. Based on this idea, we set the seventh-control hypothesis:

**Hypothesis 7 (H7).** *Gap between ICT capacities and the degree of their exploitation inhibits the intelligent use of ICT.*

### 2.1.5. Dependent Variable

The dependent variable is of primary interest to this research study. Namely, our goal is to understand and describe it and explain its variability. In the considered model, it explains to what extent the examined maritime organizations are savvy ICT exploiters. We assessed it through a questionnaire, and the eighth hypothesis in the model covers it:

**Hypothesis 8 (H8).** *Efficient and smooth communication between tasks, technologies, and employees strengthens the intelligent use of ICT.*

The research framework is shown in Figure 3. Through this methodological framework and applied methodology presented in the next section, we tested the previously set hypothesis and opened a space for further discussion among scholars and professionals in maritime administrative and business domains to increase digitalization in compliance with related sustainability goals.

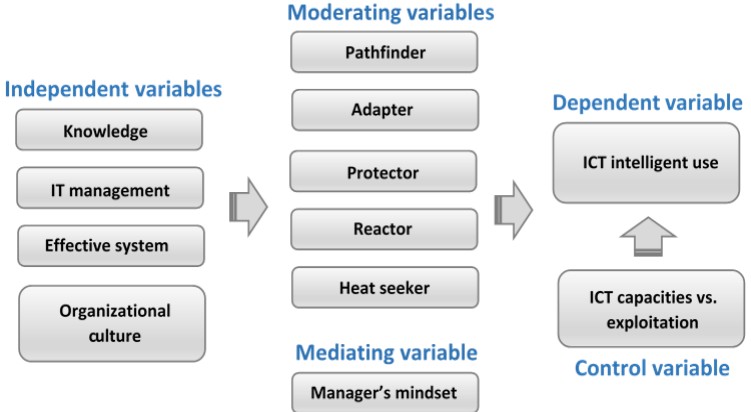

**Figure 3.** Methodological framework (Source: Own).

Concerned constructs and conjecture relationships among them directly relate to the social dimension of sustainability, but they are intertwined with environmental and economic aspects. Through social awareness and responsibility, maritime organizations are required to attain higher economic and environmental standards. However, to be profitable and reduce adverse effects on the environment at the same time has been a challenge to most maritime organizations. Tangible actions in improving environmental performance while addressing customers and society's economic and social interests will lead to truly sustainable outcomes [50]. Rational exploitation of available ICT resources in maritime and wise planning in adopting new ones are of key importance. This should include internal and external needs and expectations, corporate social responsibility, marine environmental protection (preventing sea pollution, accidents, trafficking, drug smuggling, etc.), along with economic wealth and growth (e.g., safe and green ports, "well covered" by ICT, will attract more freight and passengers, provide better operational performances, etc.). Intelligently used ICT in maritime undoubtedly supports green supply chain management and sustainable business performances.

### 3. Applied Methodology and Obtained Results

For the primary data collection, we used a structured interview. Structured interviews are widely used methods of collecting data to obtain information on an issue of interest. We have structured the interviews upon the methodological framework in the form of a questionnaire. The questionnaire contained the purpose of the interview, a set of questions in a logical order and accordance with the theoretical model, and suggested probing questions. Since we used computer-assisted interviewing, we sent the questionnaires to the interviewees via mail and reminded them twice to send us feedback in due course. In the end, we had forty conscientiously populated questionnaires by the experts employed

in maritime administrative and business companies located in four EU countries (18 interviewees): Croatia (5), Greece (6), Italy (4), and Slovenia (3); and four non-EU countries (22 interviewees): Albania (6), Bosnia and Herzegovina (5), Montenegro (10) and Serbia (1). They preferred to remain anonymous. We used a quantitative approach in terms of the Liker's interval scale with denominators 1–5 associated by linguistic labels "not important-extremely important", etc., that interviewees used to respond. The questionnaire also included a table that listed commonly used and advanced info-communication systems in maritime. Interviewees were required to mark those systems they used. In this segment, a binary approach was applied, i.e., if a system was marked as available, we anchored it by numerical value "1" and in the opposite case by a blank. Key research questions were developed based on the theoretical framework and given in Table 1.

**Table 1.** The structured interview content (Source: Own).

| Construct | | Research Question |
|---|---|---|
| C1: Knowledge | 1. | To what extent is knowledge important for business success? |
| | 2. | To what extent are the knowledge and skills of employees important for efficient and effective use of ICT? |
| C2: IT management | 3. | To what extent are ICT systems and functions important for the successful functioning of the organization and its business success? |
| | 4. | To what extent do you use ICT for operational tasks within your organization (e.g., accounting operations, a database of employees, a database of business partners, etc.)? |
| C3: Communications | 5. | To what extent can your customers use the ICT resources of your organization (e.g., your web site, various online users' applications, etc.)? |
| | 6. | To which extent does ICT allow you to become familiar with the current market trends in the area of your business? |
| C4: Organizational culture | 7. | How much is positive organizational culture and clime important for the effective use of ICT? |
| C5: Pathfinder | 8. | To what extent is the introduction of new ICT solutions risky for the organization? |
| C6: Adapter | 9. | To what extent is it important to carefully analyze existing ICT solutions prior to their introduction into the organization? |
| C7: Protector | 10. | To what extent do ICT solutions reduce operational costs of the organization? |
| C8: Reactor | 11. | To what extent can the existing ICT solutions be adapted to the current business needs of your organization? |
| C9: Heat seeker | 12. | Are the latest ICT solutions also the best ones? |
| C10: Manager's mindset | 13. | How is the manager's mindset important for the effective use of ICT? |
| C11: ICT capacities vs. exploitation | 14. | To what extent is there a divergence between ICT capacities and their real application on a daily basis in your organization? |
| C12: ICT intelligent exploitation | 15. | To what extent do ICT serve as connective tissue among tasks, technologies, and employees in your organization? |

Toward encouraging interviewees to identify their current positions and strategic trajectories for the business and the appropriate ICT strategy, constructs: Pathfinder (C5), Adapter (C6), Protector (C7), Reactor (C8), and Heat seeker (C9) were examined. All interviewees expressed their opinion regarding this issue towards the Adapter strategy. It is interesting that interviewees from both EU and non-EU countries share the same or similar attitudes towards ICT strategy orientation and business trajectories of the organizations in which they work. Only the interviewee from Serbia answers indicated equal inclinations towards Reactor and Adapter strategies (Table 2). The strategic orientation of an Adapter works in two business domains. One relatively traditional and relational, being concerned with internal controls. The other where ICT plays an increasingly important role in linking the organization to its customers or the users' marketplace. The Adapters are looking for a middle path regarding business and ICT strategic orientation. When it comes to a Reactor, this organization significantly lags others in ICT introduction. According to Stace et al. [47], in the case of Reactor, technology and ICT applications are often introduced towards the end of the technology life cycle and usually only when forced to do so by competitor pressure.

**Table 2.** The Information and Communication Technologies (ITC) strategic orientation of the interviewees (Source: Own).

| EU Respondents | ICT Strategy | Non-EU Respondents | ICT Strategy |
|:---:|:---:|:---:|:---:|
| Albania | 🟠 | Croatia | 🟠 |
| Bosnia & Herzegovina | 🟠 | Greece | 🟠 |
| Montenegro | 🟠 | Italy | 🟠 |
| Serbia | 🟣 🟠 | Slovenia | 🟠 |
| Albania | 🟠 | | |

Legend: 🟠 Adapter; 🟣 Reactor.

Mean values of the constructs: knowledge (C1), IT management (C2), effective system (C3), organizational culture (C4), managerial mindset (C10), and ICT capacities vs. exploitation (C11) are given in Table 3. It is obvious that all participants from both EU and non-EU maritime organizations highly appreciated knowledge, proper deployment of skills and roles mediated by top management, system effectiveness achieved via proper communications within the organization, with customers, and for the purpose of market exploring, as well as organizational culture highly. The control variable in all cases corresponded well to the level of intelligent use of ICT in that, in all cases, it had a relatively low level.

The interviewees from both EU and non-EU countries estimated the level of the rationality of using ICT within maritime organizations in which they work as relatively high. The results are given in Table 4. Consequently, all hypotheses in the model (H1–H8) were approved.

The obtained results were based on subjective assessments. However, when it comes to the number of ordinary and advanced ICT solutions employed [51] in the examined organizations, it became evident that maritime organizations in EU countries have considerably more ICT systems than those in the non-EU countries. The evidence is given in Table 5.

**Table 3.** The assessment of the independent, mediating, and control variables (Source: Own).

| Country/Criteria | C1 | C2 | C3 | C4 | C10 | C11 |
|---|---|---|---|---|---|---|
| **EU respondents** | | | | | | |
| **Albania** | 4.10 | 4.25 | 3.75 | 3.83 | 3.50 | 2.50 |
| **Bosnia & Herzegovina** | 4.50 | 4.00 | 3.00 | 3.00 | 3.00 | 3.00 |
| **Montenegro** | 4.40 | 4.90 | 4.80 | 4.90 | 4.80 | 2.50 |
| **Serbia** | 5.00 | 5.00 | 4.00 | 5.00 | 3.00 | 2.00 |
| **Non-EU respondents** | | | | | | |
| **Croatia** | 4.80 | 4.75 | 3.45 | 4.40 | 4.40 | 2.10 |
| **Greece** | 4.40 | 4.30 | 3.70 | 3.40 | 3.10 | 2.90 |
| **Italy** | 4.38 | 4.50 | 3.50 | 3.50 | 3.75 | 2.75 |
| **Slovenia** | 4.67 | 4.50 | 3.83 | 3.33 | 4.00 | 3.33 |

Legend: C1-Knowledge; C2-ICT management; C3-Effective system; C4-Organizational culture; C10-Managerial mindset (mediating variable); C11-ICT capacities vs. exploitation (control variable).

**Table 4.** The assessment of the level of intelligent use of ICT (Source: Own).

| Non-EU Respondents | ICT Intelligent Use | EU Respondents | ICT Intelligent Use |
|---|---|---|---|
| **Albania** | 🟡 | **Croatia** | 🟢 |
| **Bosnia & Herzegovina** | 🟡 | **Greece** | 🟢 |
| **Montenegro** | 🟢 | **Italy** | 🟡 |
| **Serbia** | 🟢 | **Slovenia** | 🟡 |

Legend: 🟡 Medium level; 🟢 High level.

**Table 5.** Availability of the ICT systems in examined maritime organizations (Source: Own).

| Info-Communication System | Non-EU Countries | | | | EU Countries | | | |
|---|---|---|---|---|---|---|---|---|
| | AL | B&H | MN | SR | CR | GR | IT | SL |
| Electronic Data Interchange (EDI) | | 1 | 1 | | 1 | 1 | 1 | 1 |
| Enterprise Resource Planning (ERP) | | 1 | 1 | | 1 | 1 | 1 | 1 |
| Customer Relationship Management (CRM) System | 1 | 1 | 1 | | 1 | 1 | 1 | 1 |
| Electronic Logistics Marketplace (ELM) | | 1 | | | 1 | 1 | 1 | |
| THETIS (PSC—Port State Control) | | | 1 | | 1 | 1 | 1 | 1 |
| Blockchain | | | | | | | | |
| Automatic Identification System (AIS) | 1 | 1 | 1 | 1 | 1 | | 1 | 1 |
| Long-range and tracking (LIRT) | 1 | | 1 | | 1 | 1 | | 1 |
| Vessel Traffic Monitoring Information System (VTMIS) | 1 | | 1 | | 1 | 1 | | |
| Sea Traffic Management (STM) | 1 | | | | 1 | | 1 | 1 |
| e-Navigation | | | | | 1 | | | 1 |
| e-Maritime | | | | | 1 | | | 1 |
| Common Maritime Communication Platform (CMSP) | | | | | 1 | | | 1 |
| Maritime Surveillance Services (MSS) | 1 | | | | 1 | | | 1 |
| 15. SafeSeaNet (SSN) | | | 1 | | 1 | 1 | | 1 |
| Maritime Single Window (MSW) | | | | | 1 | 1 | 1 | 1 |
| Automatic Guided Vehicles (AGV) | | | | | | | | |
| Digital twins | | | | | | | 1 | |
| Remotely controlled vessels | | | | | | | | |
| Unmanned area, sea or underwater vessels (UxVs) | | | | | | 1 | 1 | |
| Earth Observation Services—SAR sensors | | | | | 1 | 1 | 1 | 1 |
| Earth Observation Services—Optical sensors | | | | | 1 | 1 | 1 | 1 |
| Satellite-based oil spill detection system at sea | | | 1 | | 1 | | | 1 |
| Oil spill prediction modeling system | | | 1 | | 1 | | 1 | 1 |
| River Information Services (RIS) | | | | 1 | | | | |
| **Score:** | **6** | **5** | **10** | **2** | **18** | **12** | **13** | **19** |

Due to the binary approach applied, where "1" means availability of the system, it is clear that maritime organizations in EU countries have a considerably larger number of available ICT systems: 19 (Slovenia—SL), 18 (Croatia—CR), 13 (Italy—IT), and 12 (Greece—GR). On the other side, the non-EU countries have a significantly lower score in this regard: 10 (Montenegro—MN), 6 (Albania—AL), 5 (Bosnia and Herzegovina—B&H), and 2 (Serbia—SR). This means that the non-EU countries have to reconsider their business development strategies and ensure funds for new ICT systems and the renewal of the existing ones. The non-EU maritime administrate and business organizations should follow actual trajectories and scenarios in efficient and effective digitalization through the available sources of relevant information [52]. The appropriate ICT systems and tools are an unavoidable part of ensuring sustainability in shipping, ports, and maritime logistics [53–55], which has to take into account maritime governance, development, and interventional plans at national, regional, and global levels. The proper use of contemporary IT/IS solutions (with subsumed "C") can contribute to increasing the visibility and industry stakeholders' understanding of the current situation in sustainability (tracking sub-standard ships, accidents, pollution, waste management, contingency plans, and measures, etc.). It can also assist them in designing appropriate managerial insights and help develop appropriate sustainable policies, along with implementation strategies and methods across maritime clusters.

*Examining the Relationship between Independent and Dependent Variables*

In an attempt to refine our quantitative analysis, we modeled a multiple linear regression functional relationship between the independent and dependent variables in the model. We used all responses from both EU and non-EU countries and obtained the results presented in Table 6. The analysis was realized in the Excel Module software embedded into Microsoft Excel on an Intel(R) Core(TM) i5 CPU@1.6 GHz and 8GB RAM PC. The simulation time was negligible, i.e., we almost instantly received the results.

**Table 6.** Multiple linear regression analysis results (Source: Own).

| Intelligent Use of ICT | Knowledge | IT Management | Effective System | Organizational Culture |
|---|---|---|---|---|
| ai (i = 1, 4) | 0.124 | 0.855 | 0.107 | 0.407 |
| b | | | −2.700 | |
| r | | | 0.799 | |
| R2 | | | 0.639 | |

Coefficients a1 to a4 (Table 6) correspond to independent variables in the model: knowledge, ICT management, effective system, and organizational culture, while b represents the regression line intercept. When independent variables are jointly regressed against the dependent variable (intelligent use of ICT) in an effort to explain the variance in it, the size of individual regression coefficients indicates how much an increase of one unit in the independent variable would affect the dependent variable, assuming that all the other variables remain unchanged. In this case, IT management has the highest impact on the dependent variable, i.e., intelligent use of ICT, then organizational culture, knowledge, and system efficiency, respectively.

The individual correlations between the independent variables and the dependent variable collapse in a regression coefficient multiple r or the square of multiple r, R-square, or $R^2$, which is the amount of variance explained in the dependent variable by the predictors. The obtained regression coefficient ($R^2 = 0.639$) indicated a strong relationship between the independent variables and dependent variable in the model. In other words, if $R^2$ is 63.9%, it means that the value of the dependent variable 63.9% depends on the independent variables in the model and 36.1% on other factors (variables) that are not included in the model. This is a high correlation between the considered dependent and independent variables in the applied research approach.

## 4. Positive Experiences from Montenegro

Montenegro, as a non-EU country, has successfully implemented several info-communication systems imposed by international organizations, such as the International Maritime Organization (IMO), International Mobile Satellite Organization (IMSO), and European Commission (EC) and European Maritime Safety Agency (EMSA).

In early 2003, the Global Maritime Distress and Safety System (GMDSS) was deployed. The GMDSS was later renewed and upgraded in 2015.

In 2009, Montenegro successfully implemented Long Range Identification and Tracking (LIRT) by establishing the National LRIT Data Centre. This system enables Montenegro to use satellite-based tracking of vessels sailing under the flag of Montenegro worldwide. The positions of all ships are updated every six hours, or more often, if necessary. The system also provides information on foreign ships, which sail towards Montenegro's ports for the purpose of search and rescue at sea.

In 2013, Montenegro successfully implemented the CleanSeaNet (CSN) system previously approved by the EC and EMSA. The CSN is the European satellite-based oil spill monitoring and vessel detection service, developed and operated by the EMSA. The service analyses images, mainly from synthetic-aperture radars and optical missions. Montenegro was the first "developing country" that has joined the CSN system.

In 2015, the Vessel Traffic Monitoring Information System (VTMIS) center was officially opened in Montenegro. The sophisticated equipment for maritime surveillance, along with providing safety and security of sea traffic, was installed in three sites along its coast. The VTMIS center has required IT/IS equipment for sharing VTMIS data with respective stakeholders from Montenegro, as well as the EU partners. The personnel was trained to work as VTMIS managers and operators, as well as technicians, to maintain the system.

The aim of Montenegro is to improve the VTMIS system in the future by adding new types of sensors and new sites to cover blank spots. In addition, concepts and systems, such as augmented reality, Sea Traffic Management (STM), Common Information Sharing Environment (CISE), National Maritime Single Window (NMSW), are considered highly for the implementation. Now, the Administration for Maritime Safety and Port Management (AMSPM) is involved in several EU research projects where systems, such as CISE, unmanned aerial and underwater systems, NMSW, and STM, are subject of further research and improvements.

## 5. Discussions

The paper examined the level of rationality in using ICT solutions in selected maritime administration and business organizations in the non-EU and EU countries to identify weak points and countermeasures. The interviews were conceived to explore the employees' strategic orientation towards ICTs, along with their attitudes towards knowledge, IT management, system's effectiveness, organizational culture, and manager's mindset, while ICT was treated as a key perpetuator of sustainability in social, environmental, and economic aspects. Throughout the qualitative analysis, the following was found:

- All interviewees were Adapters when it comes to ICT strategical orientation. This means that emphasis was often on modifying rather than fundamentally re-configuring existing systems and applications;
- Only one respondent hesitated between the Adapter and Reactor. The latest meant that technology was not seen as strategic, with ICT platforms appearing weak and obsolescent.
- All respondents highly appreciated knowledge, the role of IT management, system effectiveness, positive organizational culture, and an open manager's mindset. The average assessments on the Likert scale (1–5) were above 3, while the majority were between 4 and 5.

- The respondents from two non-EU and two EU countries assessed the use of ICT in their organizations as highly intelligent, while the rest respondents assessed this key construct in the model as medium intelligent.
- Control variable in the model, which considers disharmony between the ICT capacities and exploitation, confirmed in all cases that responses negatively correspond to the level of intelligence in deploying ICTs. On the Likert scale (1–5), the control variable was between 2 and 3 for the corresponding values.
- Furthermore, there was a strong linear correlation between dependent and independent variables in the model. The value of the dependent variable of more than 60% depended on the independent variables in the model.

Through these analyses, the hypotheses (H1–H8) have been confirmed. However, when we came to the point of exploring the availability of contemporary maritime ICT systems in selected organizations, we faced the challenge of the lack of some key ICT systems for ensuring business success and sustainable development in the non-EU maritime environments. For instance, none of the non-EU examined maritime entities had strategies and plans for implementing e-Navigation, e-Maritime, maritime Cloud or CMSP, and blockchain technologies. This can be perceived as a discrepancy in implementing and adopting new strategies to increase safety and reduce the environmental impact in the marine ecosystem, which should function as unique since it is connected to the world oceans. This is the point where the EU, through its positive politics and practices, should provide recommendations and support to the non-EU maritime organizations to a larger extent.

Some positive practices in Montenegro, as a non-EU country, have been highlighted and further investigations should be oriented towards transferring these practices and positive experiences into other non-EU countries involved in sea/water transportation, in addition to the support that should be provided from the EU, at both strategical and operational levels.

## 6. Conclusions

This research has been based on the premise that the wise use of ICT in maritime enhances social, environmental, and economic dimensions of sustainability. Towards examining this premise, the employees in maritime administrative and business organizations in several EU and non-EU countries were interviewed. They had similar opinions regarding analyzed constructs connected to the intelligent use of contemporary ICT systems. Namely, they recognized knowledge, IT management, system efficiency, organizational culture, and manager's mindset as key enablers of rational and profitable use of ICT. It speaks in favor of their solid education and awareness about the importance of ICT systems and the environment in which these functions. When it came to ICT strategical orientation, they were cautious, i.e., not so prone toward taking risks in terms of investing in new ICT solutions and experimenting in the market. The applied multiple linear regression model confirmed a strong correlation between dependent and independent variables in the model. Notwithstanding, when it came to the availability of common and advanced ICT systems in the considered maritime organizations, it was shown that there were big differences between the EU and non-EU countries. For instance, maritime organizations in Slovenia had almost all the ICT systems listed in the questionnaire except blockchain or distributed ledger technology, AGVs, digital twins, UxVs, and RIS. Maritime companies in Italy had, for instance, digital twins and UxVs. Croatia and Greece also had quite extensive list of available ICT systems. On another side, examined non-EU countries were modestly equipped. The question is: what are the reasons for such differences, and how these can be addressed to avoid, in terms of sustainability, the lack of (corporate) social responsibility, disruptions in the maritime ecosystem, and negative economic implications in the future for the non-EU countries and their neighbors? This should be the subject of the following investigation in the field. In an attempt to conclude our study "optimistically", we presented some achievements that Montenegro has made regarding improvements in its maritime info-communication supra-structure. These include the renewal of GMDSS equipment ashore, implementation of LIRT, CSN, and VTMIS, working on the

introduction of STM, CISE, and NMSW, along with the consideration of opportunities for deploying sea surface and underwater unmanned vehicles for environmental safety and security missions. However, it is evident that the EU supports Montenegro in ICT for environmental and safety purposes, but business-oriented ICT that can support economic growth and development are yet not in focus. Montenegro and other examined non-EU countries should make greater efforts in this regard in cooperation with EU countries.

It is expected that non-EU countries should follow positive practices from Montenegro or other developing countries to improve their maritime ICT systems' efficiency and effectiveness. This will support ensuring sustainability in maritime, in the region, and more widely, through smart adoption of contemporary ICT solutions in cohesion with all relevant examined constructs. Further research in the field should be oriented toward policies and recommendations for maritime organizations' digitalization management, including the needs of sustainable multi- and synchro-modal transport and logistics.

**Author Contributions:** N.K. gave substantial contribution to the conception of the work, investigation, acquisition, analysis and interpretation of data for the work. S.B. is a supervisor and was responsible for conception and design of the work, methodological framework and visualization, including software analysis and results validation. I.E.D. is a co-supervisor and was responsible for editing and technical correction of the manuscript, as well as sourcing for the article processing charge. All authors have read and agreed to the published version of the manuscript.

**Funding:** Durban University of Technology funded the APC.

**Conflicts of Interest:** The authors declare no conflict of interest.

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
