# Peer review of "Digitalization in Developing Maritime Business Environments towards Ensuring Sustainability"

_sustainability, doi:10.3390/su12219235_

Round 1
Reviewer 1 Report
Dear authors,
Thank you for this interesting manuscript dealing with digitalization in european context and more specifically in developing economies contexts.
The overall manuscript is interesting and the topic in also of interest. We believe however the study does not exactly fit the scope of the journal. The study looks at digitalization of the maritime business environment, but there is not clear link presented, nor investigation of such link, between IT development and sustainability in the maritime context. The manuscript could fit better in maritime, supply chain or IT journals. In that sense, the last part of the title 'towards ensuring sustainbility' is not treated in the paper.
Please see: Sislian, L., & Jaegler, A. (2018; 2020), Fedi et al. (2019), Yuen et al. (2017), Psaraftis (2019), Lavissiere et al. (2019), Acciaro (2013, 2015), etc...
Despite the positionning of the paper, the manuscript also contains some ways of improvement. First of all, the introduction is not very sharp. There is a collection of ideas that do not flow together. Context lacks of references. For instance third paragraph on the paradox of maritime industry lacking of digitalization compared to other industries is not supported by any evidence (figures? sources? literature?). In the same way, the presentation of digitalization of maritime industry is missing. What are the areas of digitalization, why dgitalizing, what are the know processes of digitalization and diffusion of innovation. As a consequence, the introduction does not lead to a sharp research question. We don't know exactly what the authors are looking for.
In the same way, page2 with methodology is quite vague while the method used later in the paper is quite straight to the point. Authors just just state clearly what they are looking for and why they chose their method and model. And to do so, again, they should justify their approch thanks to literature. A detail: last sentence of first paragrap of methodology does not sound very scientific: "Besides, we used some other references (...) as inspiration for designing our own methodological framework". Author should say clearly what they took, where and why. Not just being inspired.
Part 2.1.1 is confusing since the authors present 3 of the elements of the model illustrated in Figure.1 (knwledge, IT maagement, effective system) plus a fourth element (organizational culture) coming from nowhere. And next part starts with the fourth element of Fig 1 (IT strategy) and its sub elements at the same leve of descritption.
In detail for this part, what are the authors bringing compared to the original model of Holtham & Courtney? why ? and on which basis ?
Also, these elements would need more sources to make robust model. It would also be interesting to have sources linking each element with maritime sector. In what are they specifically important in maritime? (I am not even mentioning anymore sustainable maritime supply chain). Hypothesis have to be more robust to be presented and tested..
To the best of the reviewers' knowledge, the mathematical model seems ok.
Conclusions are interesting, although they do not mention sustainability at all.
Author Response
First, we would like to thank anonymous reviewer for his/her valuable inputs that helped us to improve our paper.
- Thank you for your guidance to the relevant sources that consider relations between IT/IS and sustainability. We took each of them into consideration, but citied those that mostly corresponds to the content of our research paper (e.g. Sislien & Jaegler (2018, 2020), Fedi (2019) and Yen et al. (2017)).
- We extended and adapted Introduction to your suggestions.
- We described in some more detail several models, which we triangulated with Holtham’s & Courtney’s model.
- We incorporated organizational culture in the text and in Figure 1, as well. Thank you for pointing out that omission.
- We have used Holtham’s & Courtney’s model as a base for creating our own methodological model, which includes identification of dependent, moderating, mediating, control and dependent variables and relations between them. We identified and connected variables based on our previous intensive literature search, knowledge and experiences in the field, which we acquired through previous research works.
- We have conceived the questionnaires due to the previously set hypothesis, so now it is not possible to change them. Connecting constructs in the model with maritime sector will require different approach from the beginning of research. We can consider this valuable and highly appreciated idea for further investigations in this domain.
- We added Discussion to the Conclusions and highlighted the importance of smart IT/IS for reducing the number of accidents at sea and in the ports, and for reducing environmental impacts, as well as supporting social and economic growth of the considered countries, in particular non-EU ones and neighboring countries, too. (in yellow)
- The revised version of the paper is enclosed.
Thank you once again for your valuable suggestions.

Reviewer 2 Report
Thank you for the opportunity to review your work. This study supports the need for revisions the level of digitalization in several developing maritime business environments.
The presentation is clear and will hopefully inspire the needed changes, but I would personally reformulate the abstract going a little deeper through your topics.
While the subject addressed in this paper is timely and important, the paper cannot be accepted in its present form because of the following reasons:
- Explain with more detail the research problem, the research goals and the context.
- I suggest that you could separate the introduction and background in two sections.
- Discussion and conclusions: according the review of the other sections of the paper, I suggest breaking this section in two sections and in the conclusions the author should pointed out the main results of the study, the limitations and some ideas for future research.
Kind Regards.
Author Response
First, we would like to thank anonymous reviewer for his/her valuable inputs that helped us to improve our paper.
- We have extended Introduction and tried to sharp research problem and the research goals.
- We have included two sub-sections into Introduction: Research problem & Sustainability matter, since we aimed to explain with more detail the research problem, and to make paper closer to the scope of the journal.
- Thank you for this valuable suggestion. We have done this. Namely, we included Discussion in which we summarized the obtained results of the study. In addition, we highlighted limitations and directions for further research work in the field. (in yellow)
- The revised version of the paper is enclosed.
Thank you once again for your valuable suggestions.
Round 2
Reviewer 1 Report
Thanks to the authors for their improvements.
The manuscript is of a better quality. The context is better described as well as the model.
The overall sustainable aspect is still light to me. It is like the authors performed an interesting study on IT systems and they added afterwards the sustainable aspect. Paragraphs (on at the end of introduction and in conclusion) that have been added to the paper help to connect the research with sustainability, but it is still too light to me modest point of view.
The second major point is more epistemological. One cannot accept the answer 6. If the model does not fit the problem, the reviewer cannot accept this one in order to free some time to the authors to prepare further investigations in this domain.
Basically the feeling of the reviewer is minimum has been done to answer the questions of the reviewer. Most of the fundamental points highlighted during first review remain in this version.
Reviewer 2 Report
The authors submitted a new version of the article following all the suggested changes. I believe that the paper can be published in its current form.
However, there are grammar errors (dicassion) and the references do not conform to the journal's standards. These changes are necessary.
Kind regards
Author Response
We would like to thank anonymous reviewer for the second review of the paper and for valuable inputs towards its improving.
We would like to thank anonymous reviewer for the second review of the paper and for the valuable inputs towards its improving.
- Grammar and spelling errors are corrected.
- Paper is adjusted to the Journal template and references are put into the required format.
Thank you indeed for your patience and highly appreciated inputs.

Round 3
Reviewer 1 Report
The paper is of a good quality now, including the introduction, literature review and model. I however, personnaly feel like the sustainable part is not supported enough. This is a fundamental disagreement with the authors, but I recognize the quality of the work presented. I would suggest to present the manuscript in another journal with a more direct perspective. (i.e. without sustainable aspect)